# Large Memory Layers with Product Keys

**Guillaume Lample**[*†], **Alexandre Sablayrolles**[*], **Marc'Aurelio Ranzato**[*],
**Ludovic Denoyer**[*†], **Hervé Jégou**[*]
{glample,asablayrolles,ranzato,denoyer,rvj}@fb.com

## Abstract

This paper introduces a structured memory which can be easily integrated into a neural network. The memory is very large by design and significantly increases the capacity of the architecture, by up to a billion parameters with a negligible computational overhead. Its design and access pattern is based on product keys, which enable fast and exact nearest neighbor search. The ability to increase the number of parameters while keeping the same computational budget lets the overall system strike a better trade-off between prediction accuracy and computation efficiency both at training and test time. This memory layer allows us to tackle very large scale language modeling tasks. In our experiments we consider a dataset with up to 30 billion words, and we plug our memory layer in a state-of-the-art transformer-based architecture. In particular, we found that a memory augmented model with only 12 layers outperforms a baseline transformer model with 24 layers, while being twice faster at inference time. We release our code for reproducibility purposes.[3]

## 1 Introduction

Neural networks are commonly employed to address many complex tasks such as machine translation [43], image classification [27] or speech recognition [16]. As more and more data becomes available for training, these networks are increasingly larger [19]. For instance, recent models both in vision [29] and in natural language processing [20, 36, 28] have more than a billion parameters. The higher-capacity enables better modeling of data like natural text or images, and it also improves generalization [41, 33]. Unfortunately, increasing capacity has led to a dramatic increase of computational complexity, both at training and inference time [20].

There is a growing interest in developing architectures with reasonable computational complexity. Recently, there has been some efforts to develop high capacity architectures that operate on a limited computational budget [40, 18]. This is well illustrated by the "On-device Visual Intelligence Challenge" [5], which specifically focuses on the complexity/accuracy trade-off for image classification.

Some researchers have attempted to increase the capacity of a network without increasing its computational complexity. Most notably, Rae et al. [37] incorporate fast nearest neighbor search within a neural network architecture to leverage large key-value layers with sparse reads and writes. Their approach relies on an external indexing structure [32], which is approximate and needs to be re-learned regularly while training the neural network to avoid a catastrophic drift.

In this work, we propose a key-value memory layer that can scale to very large sizes while keeping exact search on the key space. This layer dramatically increases the capacity of the overall system for a negligible computational overhead. Unlike existing models based on key-value memories (see

---

[*]Facebook AI Research

[†]Sorbonne Universités, UPMC Univ Paris 06, UMR 7606, LIP6

[3]https://github.com/facebookresearch/XLM

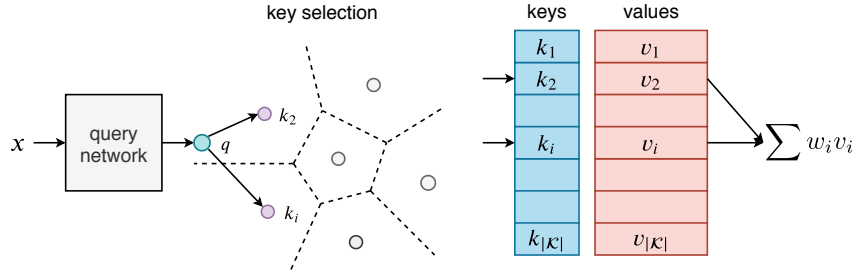

Figure 1: **Overview of a key-value memory layer:** The input $x$ is processed through a query network that produces a query vector $q$, which is compared to all the keys. The output is the sparse weighted sum over the memories associated with the selected keys. For a large number of keys $|\mathcal{K}|$, the key selection procedure becomes too expensive in practice. Our product key method is exact and makes this search process very fast.

Figure 1), we define keys as the concatenation of two sub-keys, in the spirit of product quantization [21]. As shown in more details in Figure 2, this structure implicitly defines a very large set of keys, each being associated with a value memory slot. The set of value vectors introduces the bulk of the parameters, as it scales quadratically with the number of sub-keys. Despite the large number of memory slots, finding the exact closest keys to the input is very efficient, typically requiring $\mathcal{O}(\sqrt{|\mathcal{K}|})$ vector comparisons, where $|\mathcal{K}|$ is the total number of memory slots. All the memory parameters are trainable, yet only a handful of memory slots are updated for each input at training time. Sparsity of key selection and parameter updates make both training and inference very efficient.

Our layer allows us to tackle problems where current architectures underfit given the vast amount of available data, or when they are too slow to work in practice. We thus focus on the language modeling task, integrating our memory within the popular transformer architecture [44]. This choice is motivated by the success of BERT [11] and GPT-2 [36], which demonstrated that increasing the capacity of large models directly translates to large improvements in language modeling, which in turn translates to better performance in both language understanding tasks [11, 46] and text generation [36]. Overall, our paper makes the following contributions:

- We introduce a new layer that provides a large capacity to a neural network for only a slight computational overhead both at train and test time.

- Our fast indexing strategy offers exact nearest neighbor search by construction, and avoids the pitfall of relying on an indexing structure that needs to be re-learned during training.

- We demonstrate our method within a large state-of-the-art transformer, composed of 24 layers of dimension 1600. Our method with 1 memory and 12 layers outperforms a 24-layer transformer while being twice faster at inference time. We show that adding more memory layers to transformers of various complexities provides systematic and significant improvements on our target task.

## 2  Related work

Different approaches have been proposed to increase the capacity of neural networks without increasing too much the computational complexity. For instance, conditional computation models aim at routing inputs into very large neural networks such that only a subset of connections and/or layers are used to process each input. Different methods have been developed like large mixture of experts [40], gating techniques [3, 12, 6] or even reinforcement learning-based approaches [10].

Another line of research is the development of memory augmented neural networks. For instance, memory-based neural layers [47, 42] are an efficient way to represent variable length inputs for complex problems such as question answering [48]. Such memories can also operate in feature space and have various reading and writing mechanisms [23, 17]. Unfortunately, these approaches scale linearly with the size of the memory which is prohibitive for very large memories. Neural cache models [15] suffer from the same scaling issues, which are circumvented by adopting approximate lookup techniques at test time [14].

Discretization techniques have been intensively studied for compressing network weights [8, 38] and/or activations [7, 38] or to accelerate inference. For instance, Gerald et al. [13] propose to map an input to a low-dimensional binary code, each code being associated with one category, thus reducing the complexity of inference by avoiding the use of a final large linear layer. Another model is proposed in [45], where the authors develop a fast locality-sensitive hashing technique to approximate the dot product between large matrices and vectors in neural networks. However, exploiting binary codes or approximate techniques at training time raises several challenges in terms of optimization, because approximate indexes are not accurate in high-dimensional spaces. In our paper, we borrow some ideas from product quantization (PQ) [21]. This is an approximate search technique that maps database vectors into compact codes. However, our goal is different: we do not build an approximate index, but rather we exploit the idea to represent a large set of key vectors by a drastically smaller number of vectors, that we update by regular back-propagation. As discussed later, the selection of the closest keys is exact and inherits from the fast neighbor search of PQ.

Our model is also related to sparsity models which have been mainly studied in the unsupervised learning setting [34, 24]. For instance, the k-sparse autoencoder [30] only keeps the k largest values in the latent representation of an auto-encoder, similar to our memory layer but without the product keys component. In *winner take all* autoencoders [31], sparsity is induced by using mini-batch statistics, while in the sparse access memory [37] reports some speed-up by both thresholding the memory to a sparse subset, and by using efficient data structures for content-based read operations. Unfortunately, the fast access to memories rely on an approximate external indexing structure [32] that has to be re-learned periodically. Our work solves this issue by fully incorporating the key selection mechanism as a network component.

The transformer network [44] is the current workhorse of Natural Language Processing (NLP): it is employed ubiquitously across a large variety of tasks. Transformers are built by stacking blocks composed of self-attention layers followed by fully connected layers (dubbed FFN), as shown in Figure 3. The components of the memory layer bear similarities to the query, key and value networks used in self-attention layers with two notable differences: the keys and values do not correspond to input tokens but are free embedding vectors, and the number of values (memory size) is very large.

## 3 Learnable product key memories

We consider the design of a function $m : \mathbb{R}^d \to \mathbb{R}^n$, that will act as a layer in a neural network. The purpose of $m$ is to offer a large capacity within a neural network.

### 3.1 Memory design

**High-level structure.** The overall structure of our memory is illustrated by Figures 1 and 2. The memory is composed of three components: a query network, a key selection module containing two sets of sub-keys, and a value lookup table. It first computes a query that is compared to the set of product keys. For each product key, it computes a score and selects the $k$ product keys with the highest scores. The scores are then used to produce an output $m(x)$ via a weighted sum over the values associated with the selected keys. All the parameters of the memory are trainable, yet only $k$ memory slots are updated for each input. The sparse selection and parameter update make both training and inference very efficient.

**Query generation: pre-processing network.** The function $q : x \mapsto q(x) \in \mathbb{R}^{d_q}$, referred to as the query network, maps the $d$-dimensional input to a latent space of dimensionality $d_q$. Typically, $q$ is a linear mapping or a multi-layer perceptron that reduces the dimensionality from $d$ to $d_q = 512$. As keys are randomly initialized, they occupy the space relatively uniformly. Adding a batch normalization layer on the top of the query network helps increasing key coverage during training. This insight is confirmed by our ablation experiments in Section 4.5.

**Standard key assignment and weighting.** Let $q(x)$ be a query and $\mathcal{T}_k$ denote the top-k operator[4]. Given a set of keys $\mathcal{K} = \{k_1, \dots, k_{|\mathcal{K}|}\}$ composed of $|\mathcal{K}|$ $d_q$-dimensional vectors, and an input $x$,

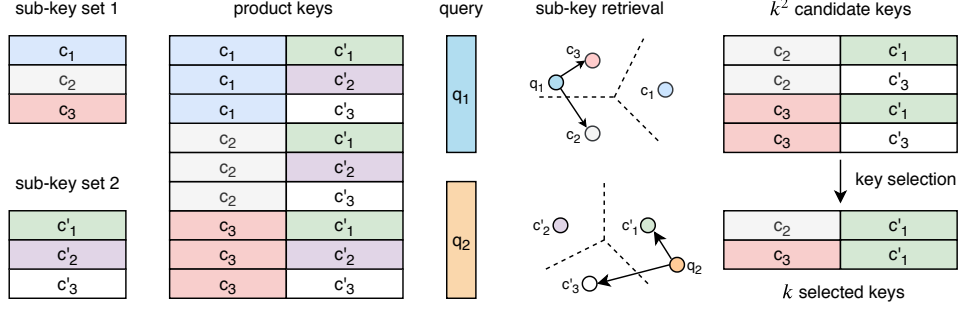

Figure 2: **Illustration of the product keys.** We define two discrete subsets of keys (sub-key set 1 and sub-key set 2). They induce a much larger set of keys, which are never made explicit (product keys). Given a query, we split it into two sub-queries ($q_1$ and $q_2$). Selecting the $k$ closest keys ($k = 2$ in the figure) in each subset implicitly selects $k \times k$ keys. The $k$ keys maximizing the inner product with the query are guaranteed to belong to this subset, on which the search can be done efficiently.

we select the top $k$ keys maximizing the inner product with the query $q(x)$:

$$\mathcal{I} = \mathcal{T}_k \left( q(x)^T k_i \right) \qquad \text{\# Get k nearest neighbors} \qquad (1)$$

$$w = \text{Softmax} \left( (q(x)^T k_i)_{i \in \mathcal{I}} \right) \qquad \text{\# Normalize top-k scores} \qquad (2)$$

$$m(x) = \sum_{i \in \mathcal{I}} w_i v_i \qquad \text{\# Aggregate selected values} \qquad (3)$$

Here $\mathcal{I}$ denotes the indices of the $k$ most similar keys (where the similarity measure is the inner product), and $w$ is the vector that represents the normalized scores associated with the selected keys. All these operations can be implemented using auto-differentiation mechanisms, making our layer pluggable at any location in a neural network.

Operations (2), (3) only depend on the top-k indices and are therefore computationally efficient. In contrast, the exhaustive comparison of Equation (1) is not efficient for large memories since it involves computing $|\mathcal{K}|$ inner products. To circumvent this issue, we resort to a structured set of keys, that we refer to as product keys.

**The product key set** is defined as the outer product, with respect to the vector concatenation operator, of two vector codebooks $\mathcal{C}$ and $\mathcal{C}'$:

$$\mathcal{K} = \{(c, c') \mid c \in \mathcal{C}, c' \in \mathcal{C}'\}$$

The total number of keys induced by this Cartesian product construction is $|\mathcal{K}| = |\mathcal{C}| \times |\mathcal{C}'|$. The sets $\mathcal{C}$ and $\mathcal{C}'$ both comprise a set of *sub-keys* of dimension $d_q/2$. We exploit this structure to compute the closest keys $\mathcal{I} \in (1, ..., K)$ efficiently. First, we split the query $q(x)$ into two sub-queries $q_1$ and $q_2$. We then compute the $k$ sub-keys in $\mathcal{C}$ (resp. $\mathcal{C}'$) closest to the sub-query $q_1$ (resp. $q_2$):

$$\mathcal{I}_\mathcal{C} = \mathcal{T}_k \left( (q_1(x)^T c_i)_{i \in \{1...|\mathcal{C}|\}} \right), \qquad \mathcal{I}_{\mathcal{C}'} = \mathcal{T}_k \left( (q_2(x)^T c'_j)_{j \in \{1...|\mathcal{C}'|\}} \right) \qquad (4)$$

We are guaranteed that the $k$ most similar keys in $\mathcal{K}$ are of the form $\{(c_i, c'_j) \mid i \in \mathcal{I}_\mathcal{C}, j \in \mathcal{I}_{\mathcal{C}'}\}$. An example of product keys with the key selection process is shown in Figure 2.

### 3.2 Complexity

Searching for the top-k most similar keys when the keys have a flat representation requires $|\mathcal{K}|$ comparisons of vectors of size $d_q$, i.e. $\mathcal{O}(|\mathcal{K}| \times d_q)$ operations.

For product keys, we consider the setup where $|\mathcal{C}| = |\mathcal{C}'|$, i.e. the configuration that maximizes $|\mathcal{C}| \times |\mathcal{C}'|$ for a fixed number of sub-keys $|\mathcal{C}| + |\mathcal{C}'|$. Since $|\mathcal{K}| = |\mathcal{C}| \times |\mathcal{C}'|$, we have $|\mathcal{C}| = \sqrt{|\mathcal{K}|}$. We only need to compare the two sub-queries with $|\mathcal{C}|$ and $|\mathcal{C}'|$ sub-keys of size $d_q/2$, which amounts to $\mathcal{O}(|\mathcal{C}| \times d_q/2 + |\mathcal{C}'| \times d_q/2) = \mathcal{O}(|\mathcal{C}| \times d_q) = \mathcal{O}(\sqrt{|\mathcal{K}|} \times d_q)$ operations.

Then, we need to search for the top-k keys in $\{(c_i, c'_j) \mid i \in \mathcal{I}_\mathcal{C}, j \in \mathcal{I}'_\mathcal{C}\}$, which is a set composed of $k^2$ keys of dimension $d_q$. This can be done in $\mathcal{O}(k^2 \times d_q)$ operations (in practice, this could be

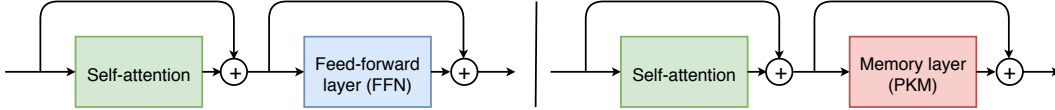

Figure 3: **Left:** A typical transformer block is composed by a self-attention layer followed by an FFN layer (a two layer network). **Right:** In our system, we replace the FFN layer with a product key memory layer, which is analogous to a sparse FFN layer with a very large hidden state. In practice, we only replace the FFN layer in $N$ layers, where typically $N \in \{0, 1, 2\}$.

done in $\mathcal{O}(k \log k)$ scalar operations with a priority list [1], but this choice is less compliant with GPU architectures). As a result, the overall complexity is:

$$\mathcal{O}\left( (\sqrt{|\mathcal{K}|} + k^2) \times d_{\mathrm{q}} \right)$$

For small values of $k$, and a memory of size $|\mathcal{K}| = 1024^2$, retrieving the nearest product keys requires about $10^3$ less operations than an exhaustive search. As shown later in our ablation study, product keys also lead to a better performance compared to a set composed of flat keys.

### 3.3   Multi-head memory attention

We make the model more expressive with a multi-head mechanism, where each head independently computes a query used to select $k$ keys from the memory. The memory simply sums the output $m_i(x)$ of each head $i$: $m(x) = \sum_{i=1}^{H} m_i(x)$ where $H$ is the number of heads.

Each head has its own query network and its own set of sub-keys, but all heads share the same values. This is similar to the multi-head attention used in transformers, except that we do not split the query into $H$ heads, but instead create $H$ queries. As the query networks are independent from each other and randomly initialized, they often map the same input to very different values of the memory. In practice, for the same input we observe very little overlap between the keys selected by two different heads. This method let us increase key usage and generally improves performance. The impact of the multi-head attention mechanism is discussed in Section 4.5.

## 4   Experiments

We report results on large-scale experiments for transformer models equipped with a memory, followed by an ablation study that shows the impact of different memory components on the model performance and memory usage. We propose to replace the FFN block of some transformer layers by a memory, as presented in Figure 3. In that setting, the memory is integrated with a residual connection in the network, and the input $x$ to the memory layer becomes $x \leftarrow x + \mathrm{PKM}(x)$ instead of $x \leftarrow x + \mathrm{FFN}(x)$. In practice, we could also keep the FFN layer and simply interleave the memory between some transformer layers.

### 4.1   Dataset

We evaluate the impact of our memory in a large scale language modeling task, where traditional models are known to underfit. The largest publicly available language modeling dataset is the One Billion Word corpus [4]. As noted in prior work [2, 9, 36], obtaining a good performance on this dataset requires tedious regularization as it is now too small for standard architectures. In our experiments, we encountered the same issues, and observed that even a small model was enough to overfit: on this dataset, for a 16 layers model with a dimensionality of 1024, we obtain a test perplexity of 25.3 when the validation perplexity starts to increase. The train perplexity is then equal to 14.8 and keeps improving while the validation perplexity deteriorates.

We therefore evaluate the benefit of our approach on a corpus that is 30 times larger and extracted from the public Common Crawl. The training set is composed of 28 billion words (140 GB of data) extracted from about 40 million English news articles indexed by Common Crawl corpora. The validation and test sets are both composed of 5000 news articles removed from the training set.

Unlike in the One Billion Word corpus, we did not shuffle sentences, allowing the model to learn long range dependencies. On this dataset, we did not observe any overfitting, and increasing the

model capacity systematically led to a better performance on the validation set. We tokenized the data using the tokenizer provided by the Moses toolkit [26]. To reduce the vocabulary size, we use fastBPE[5] to apply Byte Pair Encoding (BPE) [39], with 60k BPE splits.

## 4.2 Evaluation metrics

We measure the performance of our models by reporting the perplexity on the test set. For models with memories, we report two different metrics to evaluate the usage:

- The memory usage that represents the fraction of accessed values: $\#\{z_i \neq 0\}$
- The KL divergence between $z$ and the uniform distribution: $\log(|\mathcal{K}|) + \sum z_i \log(z_i)$

where $z = z'/\|z'\|_1$, and $z' \in \mathbb{R}^{|\mathcal{K}|}$ is defined as $z'_i = \sum_x w(x)_i$ where $w(x)$ represents the weights of the keys accessed in the memory when the network is fed with an input $x$ from the test set (i.e., the $w(x)$ are sparse with at most $H \times k$ non-zero elements).

At test time, we expect the model to access as many keys as possible, i.e. to have a usage near 100%; a lower usage means that part of the capacity is not exploited at all. The KL divergence reflects imbalance in the access patterns to the memory: if the model attends the same key for every query (while giving a tiny weight to the remaining keys), it would give a perfect usage but a very high KL, showing that the same performance could be achieved with just one value.

## 4.3 Training details

We use a transformer architecture with 16 attention heads and learned positional embeddings. We consider models with 12, 16 or 24 layers, with either 1024 or 1600 dimensions. We train our models with the Adam optimizer [25], with a learning rate of $2.5 \times 10^{-4}$, with $\beta_1 = 0.9$, $\beta_2 = 0.98$, following the learning rate schedule of Vaswani et al. [44]. In the memory, the keys and the query network are learned with the same optimizer and learning rate as the rest of the network. Since the memory values are learned with sparse updates, we found it beneficial to learn them with a higher Adam learning rate of $10^{-3}$. We implement our models with PyTorch [35], and train them on 32 Volta GPUs. We use float16 operations to speed up training and to reduce the GPU memory usage of our models. To retrieve key indices efficiently, we perform the search over sub-keys with a fast nearest neighbors implementation by Johnson et al. [22].

For a transformer model with $L$ layers and $N$ memories, we interspersed the memories at regular intervals. For instance, for $L = 16$ and $N = 2$, we replace the FFN of layers 6 and 12. This way, the network can leverage information at different levels of the architecture. The impact of the memory position within the network is studied in Section 4.5. In our main experiments, we use $H = 4$ memory heads, we select $k = 32$ keys per head, and use $|\mathcal{K}| = 512^2$ memory slots.

## 4.4 Results

| Dimension | 1024 | | | | 1600 | |
|---|---|---|---|---|---|---|
| N memories | 0 | 1 | 2 | 3 | 0 | 1 |
| 12 layers | 17.7 | 15.6 | 14.8 | 14.5 | 15.0 | 13.7 |
| 16 layers | 16.7 | 14.9 | **14.1** | - | 14.4 | **13.2** |
| 24 layers | 16.0 | 14.6 | - | - | 14.0 | - |

Table 1: **Test perplexity for models with and without memory.** PKM models with 12 layers outperforms 24-layer models of same dimensionality. Bold refers to models optimizing performance for a given dimension.

Table 1 and Figure 4 show the perplexity of different models on the test set of the CC-News corpus. We observe that increasing either the dimensionality or the number of layers leads to significant perplexity improvements in all the models. However, adding a memory to the model is more beneficial than increasing the number of layers; for instance, a model with a single memory and 12 layers outperforms a memoryless model with the same hidden dimension and 24 layers, both when the number of hidden units is 1024 and 1600. Adding 2 or 3 memory layers further improves performance.

Figure 4 also shows speed as measured in words per second, for different model configurations. In particular, when the internal hidden states have 1024 dimensions, a model with 12 layers and a

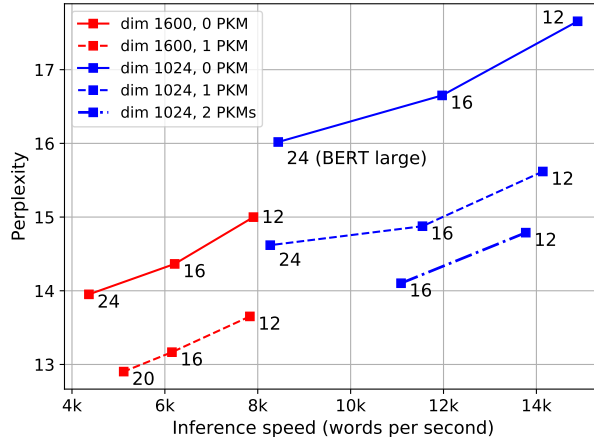

Figure 4: **Trade-off between speed and perplexity on the test set.** Labels on the graph represent the number of layers. Adding memory layers significantly improves the performance and has a negligible impact on the inference speed. Models with 12 layers and a Product Key Memory (PKM) outperform 24-layer models of the same dimension, while being almost twice faster at inference. In particular, a 12-layer model of dimension 1024 with a memory outperforms a model of 24 layers of the same dimension (same configuration as BERT large).

memory obtains a better perplexity than a model with 24 layers (same configuration as BERT large), and it is almost twice faster. When adding memory to large models that have internal dimensionality equal to 1600, inference time barely increases.

## 4.5   Ablation Study

In this section we study the impact of the different components on the memory layer, and measure how they affect the model performance and the memory usage. For all experiments, we consider a transformer network with 6 layers and 8 heads. Unless specified otherwise, we consider a memory of $512^2 = 262$k slots, with 4 memory heads, $k = 32$ selected keys, and we insert it at layer 5.

**Memory size.**   We train transformer models with memories of size $|\mathcal{K}| = |\mathcal{C}| \times |\mathcal{C}'|$, with $|\mathcal{C}'| = |\mathcal{C}|$ and $|\mathcal{C}| \in \{128, 256, 384, 512, 768, 1024\}$. Table 2 shows that test perplexity decreases as the memory becomes larger. A model with a memory size of 16k obtains a perplexity of 22.8. Increasing the size to 1M decreases the perplexity down to 18.0 while leaving the inference time unchanged. The dominant factor for inference time is the number of accessed memory values, which is governed by the number of memory heads and the parameter k, but not the memory size.

**Query Batch Normalization.**   Table 2 and Figure 5 present results with and without batch normalization in the query network. We observe that for small memories the usage is always close to 100%, but for a memory of size 1M, the batch normalization layer improves usage from 25.8% to 80.3%, with a consequent perplexity decrease from 19.8 down to 18.0. For comparison, a model without memory obtains a perplexity of 23.0, which is on par with a memory of size 16k.

Finally, we observe a correlation between the number of used keys and the model performance. In particular, a model with a memory of size 1M that does not use batch normalization uses about 25.8% of the memory values (i.e. roughly 250k values), and obtains a perplexity of 19.8, which is on par with the model using a memory of size 262k that uses batch normalization, and that has a nearly optimal memory usage of 100%.

**Memory position.**   In this experiment we insert the memory at different levels in the transformer, to see where it is the most beneficial. In Table 3 we observe that the model benefits the most from the memory when it replaces the FFN of the layers 4 or 5 in the transformer. Putting memory at layer 1 (after the input token embeddings) gives the worst performance. When the memory is inserted in layer 6, it is located right before the softmax output, the model has only one linear layer to process

Table 2: **Perplexity and memory usage for different memory sizes, with and without Batch-Norm.** Adding a batch normalization layer in the query network encourages the model to use more keys. This is not necessary for small memories of size 16k and 65k where the usage is already close to 100% without batch normalization, but for memories of size 147k of more, batch normalization improves the memory usage significantly, along with the perplexity.

| Memory size | 16k | | 65k | | 147k | | 262k | | 590k | | 1M | |
|---|---|---|---|---|---|---|---|---|---|---|---|---|
| BatchNorm | No | Yes | No | Yes | No | Yes | No | Yes | No | Yes | No | Yes |
| Perplexity | 22.8 | 23.0 | 21.7 | 21.9 | 20.9 | 20.7 | 20.5 | 19.8 | 20.0 | 18.7 | 19.8 | **18.0** |
| Usage (%) | **100** | **100** | 99.0 | **100.0** | 83.8 | 99.6 | 64.4 | 97.9 | 38.0 | 90.3 | 25.8 | 80.3 |
| KL | **0.56** | **0.56** | 0.69 | 0.58 | 0.94 | 0.65 | 1.20 | 0.68 | 1.70 | 0.83 | 2.06 | 0.95 |

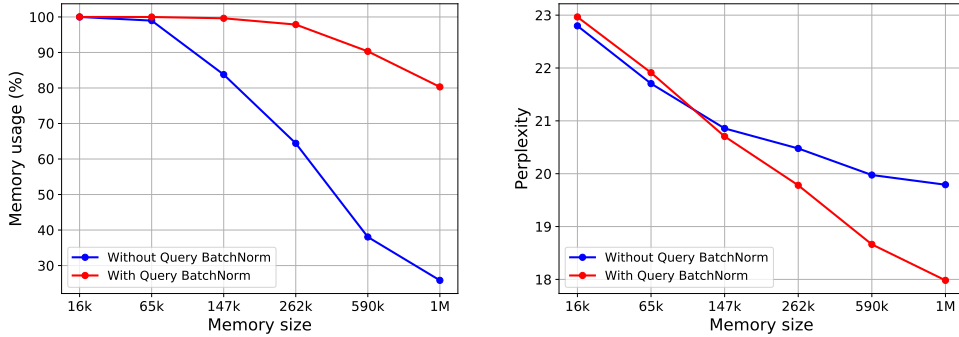

Figure 5: **Memory usage and perplexity** with and without query batch normalization. Adding batch normalization increases both performance and the fraction of used memory slots.

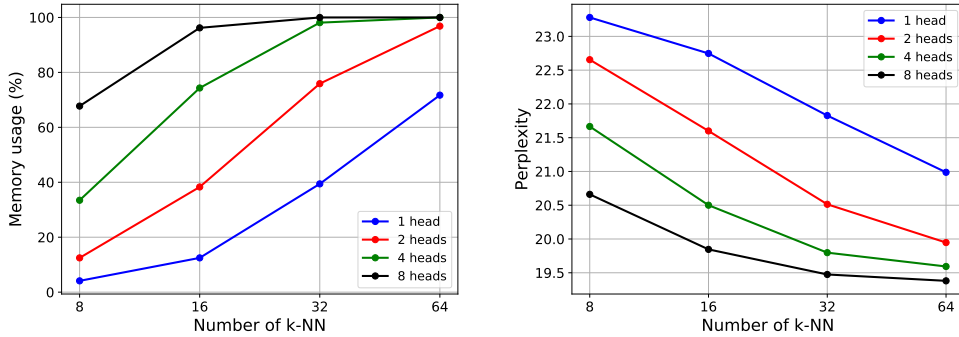

Figure 6: **Memory usage and perplexity** for different number of heads, and number of k-NN. Increasing the number of heads or k-NN increases both performance and the fraction of used memory slots.

the information read from the memory. The best position to insert the memory is at an intermediate layer. We surmise that effective use of the memory requires operating in a more abstract feature space than the input and that it is important to have some layers on the top of the memory to further process and aggregate information from every location.

**Number of heads / k-NN.** Figure 6 shows that increasing the number of heads or the number of k-NN improves both the perplexity of the model, and the memory usage. We also note that models with identical $h \times k$ ($h$ being the number of heads and $k$ the number of nearest neighbors) have a similar memory usage, i.e. models with $(h, k) \in \{(1, 64), (2, 32), (4, 16), (8, 8)\}$ all have a memory usage around 70%, and a perplexity around 20.5. Adding more heads overall improves the performance, but also increases the computation time. Overall, we found that using 4 heads and 32 k-NN strikes a good trade-off between speed and performance.

Table 3: **Perplexity and memory usage for different memory positions in a transformer with 6 layers.** Adding a memory in positions 4 or 5 maximizes the performance (layer 1 is the worst).

| Position | 1 | 2 | 3 | 4 | 5 | 6 |
|---|---|---|---|---|---|---|
| Perplexity | 21.5 | 20.7 | 20.4 | 20.1 | **19.8** | 20.3 |
| Usage (%) | **100.0** | **100.0** | 98.3 | 97.1 | 97.9 | 96.9 |
| KL | 2.23 | 0.95 | 0.74 | 0.71 | **0.68** | 1.08 |

Table 4: **Perplexity, memory usage and inference speed with product keys and regular keys.** Models with product keys have a much better usage than models that represent keys by a flat matrix, and obtain a better perplexity. They also have significantly less parameters and are dramatically faster to run. The speed is measured at inference, in thousands of words per second (w/s). For models with more than 262k memory slots, we only report the inference time. We observe that with product keys, the memory size do not impact the inference time.

| Memory size | 16k | | 65k | | 147k | | 262k | | 590k | | 1M | |
|---|---|---|---|---|---|---|---|---|---|---|---|---|
| Product Keys | No | Yes | No | Yes | No | Yes | No | Yes | No | Yes | No | Yes |
| Perplexity | 23.2 | 23.0 | 22.6 | 21.9 | 22.1 | 20.7 | - | 19.8 | - | 18.7 | - | 18.0 |
| Usage (%) | 19.6 | 100 | 13.6 | 100.0 | 10.1 | 99.6 | - | 97.9 | - | 90.3 | - | 80.3 |
| KL | 2.04 | 0.56 | 2.48 | 0.58 | 2.77 | 0.65 | - | 0.68 | - | 0.83 | - | 0.95 |
| Speed (w/s) | **35.0k** | **35.8k** | 28.5k | **36.7k** | 13.9k | **36.4k** | 7.7k | **36.3k** | 4.7k | **36.2k** | 1.2k | **35.7k** |

**Product keys vs. flat keys.** Product keys presented in Figure 2 enable finding the nearest neighbors in a matrix of size $(|C|^2, d_k)$ with the same time/compute complexity of a search over two matrices of size $(|C|, \frac{d_k}{2})$. As a result, product keys contain $|C|$ times less parameters than keys represented by a full matrix. Table 4 and Figure 7 compare product keys to the default regular flat keys. In the second case, searching the nearest keys boils down to a liner index search at each iteration, which is computationally very expensive. As a result, we only report results for memories of size 16k, 65k, 147k, as experiments with a flat index on larger memories takes an unreasonable amount of time to converge. We can see that models with product keys are not only faster but they have also a much better memory usage, and consequently obtain a better perplexity.

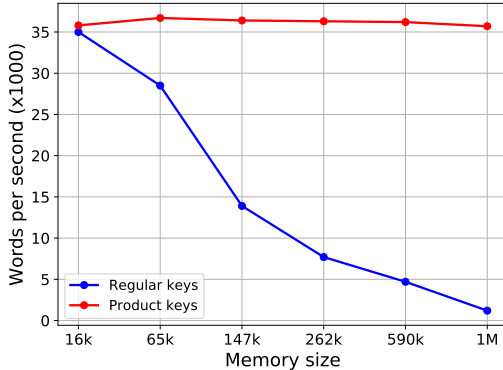

Figure 7: **Speed over memory size.** Speed (in thousands of words per second) for different memory sizes. For regular flat keys, increasing the number of keys significantly slows down the model, while with product keys, increasing the memory size barely impacts the inference speed.

## 5 Conclusion

This paper introduces a memory layer that allows to drastically improve the capacity of a neural network with a negligible computational overhead. The efficiency of our layer relies on two key ingredients: the factorization of keys as a product set, and the sparse read/write accesses to the memory values. Our layer is integrated into an existing neural network architecture. We show experimentally that it provides important gains on large-scale language modeling, reaching with 12 layers the performance of a 24-layer BERT-large model with half the running time.

## Footnotes

[4]If the permutation $(i_1, \dots, i_n)$ sorts numbers $(t_1, \dots, t_n)$ as $t_{i_1} \geq t_{i_2} \geq \dots \geq t_{i_n}$, the top-k indices are $\mathcal{T}_k(t_1, \dots, t_n) = \{i_1, \dots, i_k\}$

[5]https://github.com/glample/fastBPE

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
