[Reviews · NeurIPS 2019]

Reviewer 1



UPDATE: Authors answered my questions, I would like to keep my score unchanged and suggest to focus on clarity of the final version. Perhaps, this is the case when I would really be interested in looking at the source code. Originality: the paper borrows the general idea of product keys from the database community, however the application to fast retrieval in neural memory systems seems quite novel to me. Quality: The core ideas of the paper are sound, however more I would appreciate more rigor in both conceptual and experimental comparison with other approaches incorporating memory to Transformer (see e.g. [1]). Another suggestion would be to discuss more the issue of potential non-uniformity of the query distribution, which indeed seems to be quite relevant. There is a recent work [2] that also uses distributed sparse memory and uses a seemingly more advanced way of improving on this than a simple batch norm. Clarity: On this front the paper can be improved, and it is my main critique of the paper. Authors do not explain clearly how are the keys and values computed. Presumably those are the same as in the original Transformer network, produced on the full sequence processed to the moment, but I could not find any confirmation of this. Expressions such as "x = x + FFN(x)" (line 160) look a bit odd. I find the modification of the multi-head attention (section 3.3) unusual and requiring some explanation. If output of the heads is summed and not concatenated, how does this modify the original Transformer? Significance: I think it is a quite neat technique that is easy to implement and it will find applications in the community. [1] Chen, Mia Xu, et al. "The Best of Both Worlds: Combining Recent Advances in Neural Machine Translation." Proceedings of the 56th Annual Meeting of the Association for Computational Linguistics (Volume 1: Long Papers). 2018. [2] Rae, Jack, Sergey Bartunov, and Timothy Lillicrap. "Meta-Learning Neural Bloom Filters." International Conference on Machine Learning. 2019.

Reviewer 2



**SUMMARY * Goal: Increase the capacity of memory in memory models while keeping attentional-access cost feasible. * Model innovation: Product-Key Memory (PKM). Use as a key for a stored memory a pair of half-sized sub-keys. Treat a query as a pair of half-sized sub-queries. There are two codebooks of N sub-keys each, and the Cartesian product of these codebooks defines the set of K = N^2 full keys for the memory. For retrieval (attention): Given a query, limit retrieval to a linear combination of the values associated with the k keys that best match the query. For each sub-query, find the closest k sub-keys (maximal dot-product). The Cartesian product of these two sets of k sub-keys must contain the k full keys with largest dot-product with the full query. * Complexity: inference is reduced from O(K*d_q) to O((sqrt(K)+k^2)*d_q). * Tested models: The contents of memory are fixed at test time. (It is like Long Term Memory, not Working Memory.) In Transformer models, the FFN is replaced with a memory-retrieval network in typically 1 layer [158]. Memory size is K = 512^2 = 262k, k = 32 selected keys [226] * Task: Language Modeling on 28B-word news articles from Common Crawl corpora. * Results: E.g., Inserting one PKM layer in a dim-1600, 16 layer Transformer drops perplexity from 14.4 to 13.2 [Table 1] "Models with 12 layers and a PKM outperform 24-layer models of the same dimension, while being almost twice faster at inference." [Fig 4] Compared to comparable models with standard flat keys, "models with product keys are not only faster but also they have a much better memory usage, and consequently obtain a better perplexity" [266]. **REVIEW *The proposal is a low-level implementational modification that, for language modeling with huge training data, does seem to enable performance improvement while lowering inference cost. The memory layer proposed is in principle insertable into any model. * It is unclear how general the improvement would be across models and tasks. * The thorough set of ablation studies provided lends support to the numerous design choices. * The results are not presented with error bars, significance levels, or other indications of variability across runs.

Reviewer 3



The paper provides a "memory layer" with a large number of parameters. The parameters are accessed by a content-based attention. The attention takes a query and finds the top-K parameter vectors with the nearest keys. The innovation is in the efficient way to find the top-K keys. The efficient search splits the key to 2 parts and searches over 2 * sqrt(N_{items}) sub-keys, instead of searching over N_{items}. This still provides exact results, because the memory holds an item for each (subKey1, subKey2) combination. The experiments are done on a dataset with 28 billions words. The extra parameters are helpful there. Comments: The paper is good in all aspects. The writing is clear and pleasant to read. The experiments convincingly show the benefits of the model at the very large dataset. The memory layer may become a standard building block for large networks. I like that conditional computation models are mentioned in the related work section. The memory layer provides a nice alternative way to increase the network capacity. Minor details: - Line 122: The usage of top-K attention weights is not fully differentiable, because the k-th attention weight changes with a discontinuity when switching from rank k+1 to k. If needed, it would be possible to design a fully differentiable output, by ensuring that the k-th attention weight is zero at the tie-break. - Under line 140: The "in practice, the cost of ..." sentence should probably be: "in theory, the cost is O(k \log k x d_q) with a priority list ...". Update: Thank you for the polite feedback. The answers were informative. I hope the paper will be accepted.

[Author Response · NeurIPS 2019]

We thank the reviewers for their thorough and insightful reviews. We first respond to a common question from Reviewers 3 and 5 regarding the information stored in the memory / memory access patterns, then address each review in turn:

**Memory access (Reviewer 3, Reviewer 5).** We tried to investigate how the memory is used, and whether we can find interesting memory access patterns. For a given memory value index, we looked at the n-grams that were the most responsible for accessing this memory index. We found that for some memory indices, the associated n-grams were nicely related to a very specific topic. However, for other memory indices (in particular, the most frequently accessed ones), the pattern of the associated n-grams was not as clear. We need to investigate more to understand how exactly the model uses the memory.

**Reviewer 2**

- For the uniformity of queries, we experimented with the Kozachenko Leonenoko estimator as a loss term to favor high-entropy distribution, but we observed that it made little difference in practice. We hypothesize that a uniform distribution is a difficult target for a distribution that approximately follows a Zipf law. We thank the reviewer for suggesting Rae et al., we will incorporate it into our related work section. They propose to reallocate rarely used memory slots, to improve memory coverage. In contrast, our product key set enables to have a very good coverage of the memory and avoid this issue.
- Chen et al. compare the theoretical aspects of the Transformer and the RNN. In RNNs they argue that the memory is the hidden state, that is context-dependent. In our work, the memory consists of a set of (context-independent) parameters. The memory is of course accessed in an input specific way, but it is static unlike in RNNs.
- The keys are determined by the product set, and the values are an embedding table. Parameters of both are learned jointly with the rest of the network. The queries are computed as in a regular transformer; we will update the paper to make it clearer.
- Our multi-head mechanism is akin to the multi-head used in the attention layer, but it is used in the memory layer. We will fix the terminology to clarify this point.

**Reviewer 3**

- Our experiments are by design very large-scale and all take a large amount of time and computational power to converge (several days on 32 GPUs). Running each of them several times would be extremely expensive. However, our observation is that these experiments are overall very stable. For some particular model configurations, we actually ran several experiments with different random initializations and found that the variance was overall extremely small across runs (with differences typically smaller than 0.1 perplexity).
- We performed similar experiments (with / without memory) with the training objective presented in BERT (masked language modeling). Our findings were very similar to the ones presented in the paper (with regular language modeling): a model with 12 layers and a memory outperforms a model with 24 layers without memory, both for models of dimensionality 1024 and 1600.
- We opted for one task in order to retain the extensive empirical analysis and rigorous methodology. In our upcoming work, we are now exploring this layer for computer vision applications.

**Reviewer 5**

- The described layer is indeed not fully differentiable because there is a discontinuity when there is a switch between the k-th and (k+1)-th nearest neighbor, and "Ensuring that the k-th attention weight is zero" as suggested is a way to address this issue. This is something we actually tried, we had a hyper-parameter to remove the k-th weight to all weights (so that each weight remains positive, and the k-th one is zero) and added an extra L1-normalization step on the updated weights (to ensure they still sum to 1). However, in practice we found that the k-th weight was always very small anyway (the smallest value of a softmax over typically 32 k-NN scores), and we did not observe any difference of results by fixing this score to exactly zero.
- Our comment regarding the priority list was indeed unclear, we will clarify it in the paper.
- In Figure 1, we plot the log-probability for different different groups of words clustered by frequency. We observe that adding the memory improves the performance on all words, particularly on rare words, which is what could be expected: the memory is useful to store rare facts or rare named entities that are usually difficult for the model to retrieve. These observations are similar to what we observed by adding more layers, but no memory. We thank the reviewer for suggesting this.

Figure 1 (zoom in for details): Cumulative log-probability (higher is better) for words with different frequency, for two models with/without memory, but otherwise identical configuration. Improvements in log-probability when adding a memory layer are higher for rare words (left) than for frequent words (right).

[Meta-Review · NeurIPS 2019]

There exists some disagreement among the reviewers of this paper. Two of the reviewers believe that the introduction of the memory layer with product keys and its incorporation into the Transformer architecture is novel, interesting, and can open doors to new usecases for efficient memory-augmented neural nets. The other reviewer believes that the memory layer is merely an implementation detail, which is not proved useful for applications other than large-scale language modeling. I believe that the contributions of the paper are significant enough to grant an acceptance, especially given how important language modeling has become in modern NLP. Furthermore, because the proposed architecture is very different from common NLP and Computer Vision architectures, I recommend acceptance as a spotlight. Please improve the clarity of the technical details and address the reviewers' comments.